# OpenReview forum: "Cross-Care: Assessing the Healthcare Implications of Pre-training Data on Language Model Bias"
_NeurIPS.cc/2024/Datasets_and_Benchmarks_Track — NeurIPS 2024 Track Datasets and Benchmarks Poster_

### Official Review · Reviewer_eTqX · 2024-07-24

**Rating:** 6
**Confidence:** 4
**Correctness:** See "opportunities for improvements"
**Clarity:** See "summary"

**Review:**

See "summary", "additional questions", and "opportunities for improvement".

**Strengths:**

- The introduction of Cross-Care as a dedicated framework for assessing biases in medical applications of LLMs is a significant contribution and the availability of all data and a data visualization tool at www.crosscare.net enhances transparency and facilitates further research.
- The paper provides a thorough analysis of how biases in pre-training data manifest in model outputs, using various architectures, sizes, and alignment methods.
- By comparing model outputs with actual epidemiological data, the study addresses the practical implications of biases in healthcare, which is critical for the application of LLMs in healthcare.

**Additional Feedback:**

#Questions for authors:

- How representative are the selected 15 diseases of the broader medical landscape?
- Have you considered exploring intersectional biases (e.g., combining race and gender) to provide a more nuanced understanding of demographic impacts?
- Are there any criteria used for selecting 10 templates?
- Can you provide more detail on how ethical issues, particularly related to privacy and data protection, were addressed in the study?

**Documentation:**

See "Limitations"

**Ethics:**

Needs clarification -- see additional feedback

**Limitations:**

- Detailed explanations for some methodological choices, such as the selection of window sizes for co-occurrence analysis and the specific alignment methods could have been explained better. Excluding NER tagging could lead to inaccuracies in keyword identification and context understanding, compromising the study's validity albeit it being discussed in the limitation section. The study uses 10
different templates to assess biases but does not sufficiently explore how variations in templates might affect the results.

**Opportunities For Improvement:**

- Line 116, the authors claim to have performed a systematic literature review for each disease listed in our dictionary, focusing on prevalence and incidence within the USA across various subgroups. However, there are well-established steps and processes for conducting and reporting a systematic review, which haven’t been adequately described in this paper. Please refer to PROSPERO
(https://www.crd.york.ac.uk/PROSPERO/) and PRISMA (http://prisma-statement.org/) for more information. Following these standards could have make the review much more meaningful.

# Minor comments:
 - The clarity and legibility of Figure 1 could have been improved. It’s not vecto graphic and when zoomed in, it becomes blurry.
 - Quotations in line 108 and 109 should be adjusted. Please check the quotations throughout the paper.

**Relation To Prior Work:**

See "opportunities for improvement"

**Summary And Contributions:**

The paper presents Cross-Care, a benchmark framework that focuses on disease prevalence representation across different demographic groups and is intended to assess biases and real-world knowledge in large language models (LLMs). The study contrasts LLM outputs with real illness prevalence data from U.S. demographic groups to examine how demographic biases in pre-training corpora such as The Pile affect LLM results. The report identifies significant misalignments between real-world data and LLM illness prevalence representations, indicating a potential for bias propagation in medical applications.

---- After Rebuttal ---
The authors have clarified some aspects of the work, and based on other reviewers' comments, I am updating my scores.

---

> ### Author Rebuttal · Authors · 2024-08-14
>
> We are grateful for the reviewer's detailed feedback and their recognition of Cross-Care's contribution and the thorough analysis of LLM biases presented. We appreciate the reviewer pointing out that our study and framework are significant contributions, transparent, and facilitate future research in the field.
>
> We acknowledge the reviewer's concerns regarding the systematic literature review, methodological explanations, and exploration of intersectional biases, and would like to clarify these points below.
>
> ### Can you clarify the meaning and application of the term Systematic Literature Review?
>
> We apologize for the confusion we may have caused by referring to our high-quality disease prevalence data collection using the words “systematic literature review.” We were trying to express that we had a reliable and reproducible process of collecting disease prevalence because there is currently no single source of disease prevalence data overall or broken down by demographic subgroup from which we could draw. A systematic review, in its meaning as a specific type of research study, of disease prevalence is a standalone biomedical research undertaking in itself and outside the scope of our research. Systematic reviews aim to review the literature to answer a research question, which is not the aim of our study. Instead, our focus was on collecting reliable prevalence data from reputable sources, primarily the CDC, to establish a preliminary benchmark for comparing model outputs using an approach that was as transparent and reproducible as possible. We aimed for a reproducible, reliable approach in our data collection, prioritizing government and international agency reports and carefully documenting our search strategy and data sources. This strategy is reported in Appendix A. We will change our language from “systematic literature review” to “standardized process” in Section 3.1.
>
> ### Can you improve the clarity of Figure 1?
>
> Yes! For the camera-ready version, we will make sure to update this figure to a vector graphic.
>
> ### Quotations in lines 108 and 109 should be adjusted. Please check the quotations throughout the paper.
>
> Thank you for noticing this. We will update the quotations for the camera-ready version.
>
> ### Can you provide clearer explanations regarding the methodological choices?
>
> Thank you for highlighting these points.
> We have specified the reason for selecting the window sizes in Section 3.1, which is shown below:
>
> > We used windows of 50-250 tokens to capture co-occurrences between disease and demographic keywords. This range was chosen based on the intuition that, if in relation to one another, disease and demographic keywords should appear within 1 sentence to a short paragraph of one another and that longer distances would tend to capture spurious co-occurrences.
>
> We also highlight the reason for excluding the NER tagging in Section 3.1, but will provide a clearer justification below:
>
> > Named entity recognition (NER) tagger methods that could aid delineation of the use of specific keywords in a specific context, e.g., “white” or “black” referring to race, versus in other use cases, e.g., “white blood cells,” were initially trialed. However, using NER became infeasible under available time and compute constraints and therefore was not used in the final analysis. This should be an active area for future work.
>
> If you feel further clarification is needed on these points, please let us know, and we will be happy to add to this.
>
> ### Did you look into how Template Variation impacted results?
>
> Yes, this finding was in Appendix B, Figures 5 and 6. We evaluated the consistency of model rankings across different template variations and the results exhibit good agreement. To address this concern, we will highlight these findings more prominently in the manuscript for the camera-ready version.
>
> ### How representative are the selected 15 diseases of the broader medical landscape?
>
> This is an important question. To maximize applicability to the broader medical landscape, we focused on diseases that had the availability of consistent, high-quality prevalence data from the CDC. While not encompassing the entire medical landscape, they include the most common chronic diseases in the US, contributing significantly to the disease burden, and the leading cause of death—cardiovascular disease. This selection offers a meaningful starting point for assessing bias. We also want to emphasize that our co-occurrence analysis covered 89 diseases, providing a broader perspective on disease representation in the pretraining data.
>
> ### Are there any criteria used for selecting 10 templates?
>
> These templates were selected to be plain representations of demographic prevalence as multi-token probabilities were used. Additional variations facilitated the exploration of sensitivities to specific combinations across models.
>
> ### Can you provide more detail on how ethical issues, particularly related to privacy and data protection, were addressed in the study?
>
> This is an important point, and we will add it to the ethics section of the checklist. All data used in this study were publicly available, and no personally identifiable information was collected or analyzed.
>
> If you have additional concerns or questions, please let us know.

---

> > ### Author Rebuttal · Authors · 2024-08-14
> >
> > ### Have you considered exploring intersectional biases (e.g., combining race and gender) to provide a more nuanced understanding of demographic impacts?
> >
> > We agree that exploring intersectional biases is crucial for more nuanced discussions on this topic. However, as mentioned above, we were limited by the availability of high-quality intersectional demographic data on medical disease prevalences. This will be highlighted in the future work section, as noted in Reviewer 2:
> >
> > #### 5.x Future Work
> >
> > > This work has highlighted a fundamental disconnect between real-world prevalence estimates and LLM outputs, which appear to > track significantly closer to simple co-occurrences in pre-training data. In order to address these discrepancies, the most > obvious solution is to curate pre-training data of language models with this knowledge in mind and for a specific context. > Furthermore, organizations and regulators can evaluate simple co-occurrences to provide a rough idea of models' tendencies > before deployment and in addition to task performance. This is particularly important when considering multilingual models; if > this is to be used across languages, then accurate data in these languages are important at both pretraining and alignment stages.
> > >
> > > Furthermore, for researchers, our work highlights exciting opportunities for leveraging our findings to drive the development of > new debiasing strategies in LLMs. In particular, we would like to highlight the following concrete problems:
> > >
> > > - Ability of RAG (Retrieval-Augmented Generation) to update prevalence estimates compared to fine-tuning methods.
> > > - Use of continued pretraining or fine-tuning with Real-World Data-Aware Synthetic Data to explicitly incorporate real-world > prevalence statistics, aiming to align model predictions with actual disease distributions.

---

> > > ### Comment · Reviewer_eTqX · 2024-08-26
> > > **Updating my scores -- thank you for your clarifications**
> > >
> > > The authors have addressed the key concerns raised in my initial review. They clarified the use of "systematic literature review", explaining it as a "standardized process," and have committed to updating the terminology. They also plan to improve the clarity of Figure 1, correct the quotation issues, and provide additional explanations for their methodological choices. Their responses on the selection of diseases and template variations were reasonable. Given these clarifications and the significant contribution of Cross-Care in evaluating LLM biases in healthcare, I am increasing my score.

---

### Official Review · Reviewer_3CK8 · 2024-07-25
**Systematic exploration of biases related to disease prevalence**

**Rating:** 8
**Confidence:** 4
**Clarity:** 1. Overall the paper is quite well wr…

**Review:**

This paper presents an analysis of the biases that may be present in LLM outputs on a diverse array of aspects - comparison between real-world knowledge and LLM generations, co-occurrence between specific demographic subgroups and disease prevalence - both in the real world as well as in training corpora, the effect of alignment methods on different trained versions of the same model, etc. The analysis presented is thus quite extensive, original and provides a clear picture of the problem at hand.

**Strengths:**

1. Analysis across multiple languages
2. Tackles an important problem - positive social impact
3. Falls under the domain of responsible AI - findings hold significant importance in the status quo
4. Analysis is quite vast and fairly exhaustive. For example, the distinction between models in the wild and controlled group - this is an important distinction to make
5. Overall workflow of crosscare (figure 1) is systematically explained and simple to understand

**Additional Feedback:**

While this paper dives deep into unveiling the existing representative biases in LLMs and the lack of grounding of their outputs in real-world medical knowledge, the true impact of this work cannot be reached until the scope of this paper goes beyond analyzing the problem to actually link to the solution. The extent of biases in LLMs are well established but concrete future steps to enhance LLMs would truly solidify this paper. What future course of action can be adopted, taking cues from the findings?

**Correctness:**

Claims made in this paper are based on sound and well-explained assumptions, following fairly rigorous experimental design.

**Documentation:**

Well-documented.

**Ethics:**

No ethical concerns.

**Limitations:**

No intended negative social impact of this work.

**Opportunities For Improvement:**

1. Figures 2, 3 and 4 are difficult to grasp, further explanation is needed.
2. Figure 3b: Can we attribute the gap between blue/green and red plots to the difference between real world data and the data points captured in The Pile?

**Relation To Prior Work:**

Adequately discussed.

**Summary And Contributions:**

The paper dives deep into uncovering representational biases in LLMs, specifically in the context of healthcare and disease spread amongst different populations. Major contributions include:
1. Co-occurrence statistics in pretraining corpus mapping demographics with disease prevalence
2. Testing how disease-demographic subgroup pairs translate to raw model outputs
3. Juxtaposition of actual disease prevalence on top of model predictions to gauge discrepancies, even across multiple languages
4. A visualisation and exploration online tool backed by these results and experiments

---

> ### Author Rebuttal · Authors · 2024-08-14
>
> We are very grateful for your comprehensive review and positive assessment of our work. We appreciate your recognition of the extensive and original analysis, the clear explanation of the problem, and the importance of our findings for responsible AI in healthcare. We are particularly pleased that you found our workflow (Figure 1) systematic and easy to understand.
>
> Each of your points for improvement is addressed below:
>
> ### Can you add more clarification to Figures 2, 3, and 4?
>
> We acknowledge that Figures 2, 3, and 4 are dense and could benefit from further clarification. We will revise these figures and provide additional explanations in the camera-ready version (leveraging the extra one page) to improve their clarity and highlight how they support our claims:
>
> - **Figure 2**: We will simplify the visual representation and include a clearer legend explaining the color coding for the ranking systems (The Pile, Llama3, and real-world prevalence). In the legend, we will also add content summarizing the key findings that these three sources of disease rankings do not align with each other.
>
>     **Figure 2**: *Comparison of disease rankings between The Pile (Blue), Llama3’s logits (Green), and real-world data (Red). Position of the marker indicates the relevant ranking of each attribute for a given disease demographic pair (1: most prevalent, 5: least prevalent). For example, looking at the disease “Perforated Ulcer,” The Pile ranked White race most prevalent, Llama3 logits second, and the real prevalence ranked third.*
>
> - **Figure 3**: We will break down this figure with a more textual walk-through, focusing on the key takeaways that our findings demonstrate evidence of decoder-only models heavily relying on their pretraining co-occurrence distribution. We will also add annotations to emphasize the discrepancies between logits and co-occurrence rankings.
>
>     **Figure 3**: *a) …  b) … The overlap of green and blue lines indicates consistency across our subset and the full 89 diseases. The gap between these two lines and the red line highlights the greater association with co-occurrences compared to real-world prevalence.*
>
> - **Figure 4**: We will refine the labeling and color scheme to improve readability and provide a more intuitive understanding of the trends across different alignment strategies. In the legend, we will also add content summarizing the key findings that (1) logit-based rankings vary across alignment strategies, and (2) alignment strategies impact logit-based rankings differently across languages.
>
>     **Figure 4**: *Top ranked gender and race/ethnicity subgroups across each of the 89 diseases and different alignments of methods for Llama2 models according to logits results (stacked bars). The change in top ranked demographic from base model to respective tuned models illustrates the varying impact of alignment strategies on downstream ranking. Note this variation with various tuning strategies is not uniform across languages.*
>
> ### Can you clarify the gap between blue/green and red plots in Figure 3b?
>
> Yes, this is an important point and your intuition is correct!
> This gap is the difference between real-world data prevalences and the co-occurrence data points captured in The Pile. This discrepancy underscores the lack of real-world grounding in the pretraining data, leading to misalignments between model predictions and disease prevalence. We will clarify this point in the main text and Figure 3b legend and emphasize that these findings provide evidence of the importance of incorporating accurate real-world data to improve the grounding of LLMs in medical knowledge.
>
> ### Are you able to link the problems you found to potential solutions?
>
> We wholeheartedly agree that analyzing the problem is only the first step.
>
> We agree that providing clear next steps for researchers and broad principles that can be adhered to is important. To address this point, we will add the following section to the final section of the camera-ready version.
>
> #### 5.x Future work
>
> >This work has highlighted a fundamental disconnect between real-world prevalence estimates and LLM outputs, which appear to track >significantly closer to simple co-occurrences in pre-training data. In order to address these discrepancies, the most obvious solution is to >curate pre-training data of language models with this knowledge in mind and for a specific context. Furthermore, organizations and >regulators can evaluate simple co-occurrences to provide a rough idea of models' tendencies before deployment and in addition to task >performance. This is particularly important when considering multilingual models; if this is to be used across languages, then accurate data >in these languages are important at both pretraining and alignment stages.
>
> Furthermore, for researchers, our work highlights exciting opportunities for leveraging our findings to drive the development of new debiasing strategies in LLMs. In particular, we would like to highlight the following concrete problems:
>
> - Ability of RAG (Retrieval-Augmented Generation) to update prevalence estimates compared to fine-tuning methods.
> - Use of continued pretraining or fine-tuning with Real-World Data-Aware Synthetic Data to explicitly incorporate real-world prevalence statistics, aiming to align model predictions with actual disease distributions.
>
> ### Can you explicitly mention the co-occurrence pipeline from previous work?
>
> Of course! We will add to our brief description to more explicitly explain key sections to alleviate the repeated references for the camera-ready version.
>
> *Let us know if you have any more concerns or further clarifications!*

---

### Official Review · Reviewer_dKg5 · 2024-07-25
**Cross-Care: Assessing the Healthcare Implications of Pre-training Data on Language Model Bias**

**Rating:** 7
**Confidence:** 3
**Correctness:** Yes

**Review:**

## Quality

**Pros**:
- **Comprehensive Analysis**: The paper provides an in-depth examination of biases in LLMs, focusing on demographic disparities. This thorough approach includes various demographic subgroups and uses robust methodologies for assessing these biases.
- **Use of Baseline Models**: The inclusion of Pythia and Mamba models, trained on The Pile dataset, strengthens the study by providing a clear assessment of biases present in the pre-training data.

**Cons**:
- **Over-Simplification of Demographics**: The study uses broad demographic categories, which might oversimplify complex identities and miss more nuanced biases.
- **Potential Errors in Translation**: The use of GPT-4 for translating English templates into Chinese, French, and Spanish may introduce errors, especially in nuanced or culturally specific content. This reliance on an automated translation tool could lead to inaccuracies in the translated templates, affecting the overall reliability of the data used for bias assessment.

## Clarity

**Pros**:
- **Clear and Structured Presentation**: The paper is well-organized, making it easy to follow the research problem, methodology, results, and implications.
- **Detailed Methodological Descriptions**: Comprehensive explanations of the methodologies used, including co-occurrence analysis and statistical measures like Kendall’s tau, aid in understanding the study's approach.

### Significance

**Pros**:
- **Impact on Healthcare AI**: The study's findings are vital for understanding and mitigating biases in healthcare-related AI applications, which can lead to more equitable healthcare delivery.
- **Resource for Future Research**: The co-occurrence data and visualization tools provided are valuable for further research in AI fairness and healthcare applications.

**Strengths:**

The paper's innovative methodology benchmarks the co-occurrence of disease and demographic keywords in large corpora like the Pile datasets against real-world prevalence data. This approach provides a robust framework for identifying and assessing biases in large language models (LLMs). One of the strengths of this method is its scalability, allowing it to be applied across various domains beyond healthcare. This scalability ensures that the methodology can be adapted to different datasets and demographic groups, making it a versatile tool for bias detection in AI applications.

**Additional Feedback:**

None

**Clarity:**

The paper is structured with well-defined sections, including a thorough introduction, detailed methodology, and comprehensive results.

**Documentation:**

Yes

**Limitations:**

Yes

**Opportunities For Improvement:**

None that I could think of

**Relation To Prior Work:**

Yes

**Summary And Contributions:**

This work offers a comprehensive analysis of systematically identifying biases in large corpora, such as The Pile, to mitigate the propagation of these biases in LLM-driven applications. By thoroughly examining demographic biases and providing detailed co-occurrence data, the study highlights the importance of grounding language models with accurate and representative datasets. The use of Pythia and Mamba models, which are trained on The Pile dataset, adds robustness to the analysis by directly assessing biases inherent in the pre-training data. This approach is crucial for developing more equitable and reliable applications of LLMs, particularly in critical areas such as healthcare.

---

> ### Author Rebuttal · Authors · 2024-08-14
>
> Thank you for the thorough review and all your thoughtful feedback. We appreciate your positive assessment of our work, particularly highlighting the comprehensive analysis, the strong framework we built for visualization and future analysis, the use of baseline models for evaluations, clear presentation, detailed methodological descriptions, and the significance of the findings for healthcare AI and future research.
>
> We address each of your concerns in turn below:
>
> ### Can you address the Over-Simplification of Demographic terms used?
>
> We agree that our use of broad demographic categories might oversimplify the complexities of individual identities and could miss more nuanced biases. Unfortunately, however, in order to ground this work in real-world prevalence estimates we were restricted to the level of granularity of the United States census data, which were used in the CDC surveys to collect disease statistics across demographic subgroups.
>
> More granular information on intersections or other characteristics are unfortunately not collected.
>
> To address this point we have adjusted Section 5 (Limitations and Future Work) to state:
>
> > Demographic categories were limited by the granularity of information available in national level statistics, these are inherently simplistic, will miss more nuanced biases such as intersectionality and may contribute to stereotyping. To address the nuanced biases facing real-world problems, it is important that this information is collected and distributed to allow future work to tackle these problems.
> >
> > This statement highlights your important point and directly points to the limiting factor preventing more nuanced evaluations and can spur future dataset curation efforts and work that people can build upon our work.
>
> Nevertheless, we believe this work provides a valuable foundation for future research that can delve deeper into the nuances of subgroup robustness, emphasizing locally designed and governed approaches to address the limitations of broad categorizations.
>
> ### Is there a risk of errors in the template translation?
>
> We acknowledge the potential for errors introduced by using GPT-4 for template translation. We mitigate this risk by having native speakers review and revise all of the translated templates. We detail this in the experimental framework description (Section 3.4). And we will adjust Section 5 (Limitations and Future Work) on this potential limitation.
>
> We are excited to release these resources to help ML and medical researchers from diverse cultural backgrounds leverage and build upon our work. We are committed to continually improving the robustness of our methods and welcome further suggestions to ensure accurate cross-lingual comparisons.
>
> *Let us know if you have any more concerns or further clarifications!*

---

### Author Response · Authors · 2024-08-14
**Overall Response**

We would like to thank all the reviewers for their thoughtful and detailed feedback on our submission. We are encouraged by the overwhelmingly positive responses! All three reviewers commended the paper for its comprehensive and original analysis of biases in large language models, particularly in the context of healthcare. The strong methodology, including the use of baseline models and the Cross-Care framework, was recognized as a significant contribution, providing a robust and scalable approach to bias detection. The paper's clear and systematic presentation, along with its impactful findings for healthcare AI, was consistently praised. Additionally, the co-occurrence data and visualization tools were valued as important resources for future research in AI fairness.

We appreciate the constructive criticism provided by Reviewer 1 and Reviewer 2, both of whom highlighted valuable areas for improvement. We have addressed these points in detail in our responses, incorporating suggestions to further clarify our figures, enhance the explanation of our methods, and explicitly link our findings to potential solutions. These revisions will strengthen the final version of our paper and enhance its overall clarity and impact.

Regarding Reviewer 3's feedback, while we acknowledge the concerns raised, we believe the identified issues are largely addressable and do not diminish the overall merit of our work. We've clarified points related to the systematic literature review, methodological choices, and exploration of intersectional biases in our response. We note that the score provided by Reviewer 3 may not fully reflect the actual critique, especially given the alignment of their feedback with the more positive assessments from other reviewers.

We hope that our detailed responses and planned revisions adequately address the concerns raised and that the paper will be considered for acceptance. We are confident that the contributions of this work will provide a strong foundation for future research in the field of AI fairness and healthcare applications.

Lastly, we emphasize that all our data and code are open source. Additionally, we developed a comprehensive visualization tool/website, www.crosscare.net, designed to enhance accessibility for both computer scientists and medical professionals.

---

### Decision · Program_Chairs · 2024-09-26

**Decision:**

Accept (Poster)

**Comment:**

The Cross-Care benchmark framework analyzes biases in language models' representations of disease prevalence across demographic groups, exposing discrepancies between model outputs and real-world data, emphasizing the need for accurate data and debiasing strategies in healthcare applications of LLMs. The paper presents a comprehensive analysis of biases in LLMs, including the use of baseline models and a visualization tool for future research in AI fairness. The authors acknowledge limitations related to the granularity of demographic data and potential errors in translation but emphasize the scalability and adaptability of their methodology. The paper receives overall positive feedback, leading to a decision to accept it.